# Studies and Considerations on Forty-Three Gold and Silver Nose Ornaments from the Moche Tomb of the Lady of Cao

Giovanni E. Gigante [1,*], Roberto Cesareo [2], Angel Bustamante [3], Arabel Fernandez [4], Régulo Franco [5], Soraia Azeredo [6] and Ricardo T. Lopes [6]

1. Department of Basic and Applied Sciences for Engineering, University of Rome "La Sapienza", 00161 Rome, Italy
2. Istituto di Matematica e Fisica, University of Sassari, 07100 Sassari, Italy; roberto.cesareo@gmail.com
3. Departamento Académico de Física Del Estado Sólido, Universidad Nacional Mayor de San Marcos, Lima 15081, Peru; angelbd1@gmail.com
4. Centro de Investigación Textil Chuguay, Trujillo 13001, Peru
5. PACEB Museo Cao and Fundación Wiese, Lima 15047, Peru; regulofrancoj@gmail.com
6. COPPE, Instituto Alberto Luiz Coimbra de Pós-Graduação e Pesquisa de Engenharia, Universidade Federal do Rio de Janeiro, Rio de Janeiro 21941-614, Brazil; soraia@lin.ufrj.br (S.A.); ricardo@lin.ufrj.br (R.T.L.)
* Correspondence: giovannie.gigante@gmail.com

**Abstract:** The authors studied forty-three beautiful nose ornaments from the Moche tomb of the Lady of Cao, located in the north of Peru, which has been dated to be around 300–400 d.C. Of these items, thirty-nine are composed of a sheet of gold alloy joined together in various manners to a silver alloy sheet, which provides a strong contrast at their interface. Two nose ornaments are on gold alloy and two on silver alloy. These nose ornaments were studied using the following methods: (i) Energy-dispersive X-ray fluorescence (EDXRF); (ii) Transmission of monoenergetic X-rays (XRT) and (iii) X-ray Radiography (RAD). The conclusion, deduced from all applied methods, was that two sheets of gold and silver alloys were joined together with various methods, including gluing, mechanically joining, soldering, smelting with the aid of heating or using mercury to create an amalgam. It cannot be excluded that a few areas, visibly appearing as silver, were obtained by depletion silvering from the base Au-Cu-Ag alloy. By analyzing a fragment from the silver area of a nose ornament and by studying a few other nose ornaments from the tomb of the Lady of Cao in situ, G. Ingo and co-workers concluded that a unique sheet of three-component alloy (Ag-Cu-Au), whichemployed and transformed the surface of the objects to appear to be gold and silver by depletion gilding and silvering.

**Keywords:** gold artifacts; silver artifacts; XRF; transmission of monoenergetic X-rays; radiography; Moche; manufacturing technique

## 1. Introduction

On the north coast of present-day Peru, the Moche civilization flourished approximately between 100 and 700 d.C. Their impressive metalworking capabilities were demonstrated when Walter Alva and co-workers discovered, in 1987, the "tumbas reales de Sipán" [1,2], and in 2005, when Régulo Franco discovered the tomb of the Lady of Cao [3,4]. Huaca Cao was a religious and the most important building in the Chicama Valley. It was also a major mausoleum because high-rank personages were buried inside, whether these were priests or Moche elite personages, such as the Lady of Cao [5]. The funerary bundle of the Lady of Cao is shown in Figure 1 and was 181 cm long by 75 cm wide and 42 cm high, with an estimated weight of 120 kg.

The appearance of precious metals in Europe in different areas, starting from the Chalcolithic period [6], is an interesting Archeometallurgical task. The first evidence of

gold metallurgy in Greece in the Bronze Age [7] and in the Italic (Etruria) and Iberian Peninsula [8,9] is an important topic for archeology in different areas.

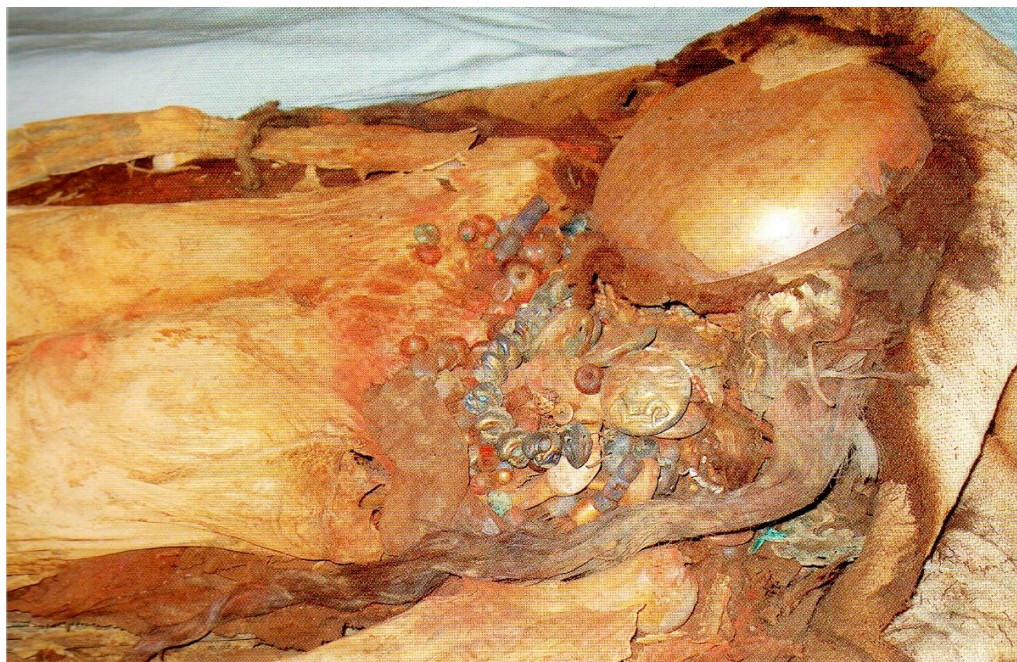

**Figure 1.** Sarcophagus and part of the body of the Lady of Cao (after Ref.). She was 1.48 meters tall, was 25 years old and died of eclampsia. The face of the Lady is covered by bowl 073, which is gilded copper, while the body is resting on 23 ceremonial spear throwers covered with gilded copper.

Spectacular gold and silver funerary ornaments were excavated and are now on display in the Museum "tumbas reales de Sipán" in Lambayeque and in the Museum of Cao, close to Magdalena de Cao, about 60 km north of Trujillo. Their condition is excellent, as there is no apparent corrosion of the surfaces. There is only an oxidation process on the silver surfaces, which requires careful maintenance.

Of the treasures of the Lady of Cao, forty-three magnificent nose ornaments stand out, the images of which are shown in the Appendix A. The nose ornaments are all similar sizes, but the shapes are different. In general, in one single nose ornament, the representations come in pairs, with the decorative motifs facing each other, on their back or superimposed, as following a dual organization. A good number of them combine gold and silver, which is a duality symbol quite common in the Moche society and essential to the Andean worldview. Silver came from the moon, a female entity, while gold came from the sun, its male opposite. Nowadays, it is difficult to understand this bimetallic duality, also because gold is about 100 times more costly than silver.

These 43 ornaments were carefully studied using the X-ray-based analytical methods of Energy-dispersive X-ray fluorescence (EDXRF), Transmission of monoenergetic X-rays (XRT) and Radiography (RAD). Along with, in a few cases, micro-tomography(μT) [10–16]. In particular, the nose ornaments were analyzed by EDXRF numerous times at various points [10], while the other methods were employed in specific cases [10,16]. The methodological approach was to examine almost all the jewels with the aim of identifying which production techniques were the prevailing technologies.

Concurrent to this work, a complete study of a few fragments from the tomb of the Lady of Cao was carried out by G. Ingo and co-workers [17]. In particular, a fragment of a silver area of the left human figure of nose decoration 105 (erroneously identified by Ingo as 003) was analyzed, and a few other nose decorations from the same tomb were analyzed in situ, i.e., nose decorations 022 and 024. While Optical Microscopy was applied to the last two objects, the following analytical methods were employed for analyzing

the fragment of nose decoration 00105: scanning electron microscopy (SEM) coupled with energy dispersive spectroscopy (EPS); X-ray diffraction (XRD); X-ray photoelectron spectroscopy (EDS); Optical Microscopy (OM). The authors conclude as follows [17]:

*The manufacturing method used by the Moche goldsmiths to chemically modify the surface of Cu-Au-Ag alloys, in some cases achieving the contemporaneous presence of gold and silver areas, was the depletion gilding.*

This assertion means that many, if not all, the nose ornaments from the tomb of the Lady of Cao could have been produced by starting with a sheet of Cu-Au-Ag alloy and manipulating its surface by depletion gilding or silvering to create gold and silver areas.

In this regard, results are presented, which clearly show that the prevailing technique was to join separate gold and silver sheets. On this aspect, the results of the radiographs are very clear (see Figures in discussion).

## 2. Depletion Gilding

Objects made of gold with copper and silver can show a natural phenomenon of surface enrichment due to a selective migration to the surface of other elements besides gold, i.e., copper and silver. This process can be artificially accelerated, and these metals (copper and silver) are then etched away from the surface by means of some acid or salt, often combined with the action of the hand. The processes of induced migration and removal of copper (and silver) from the surface are repeated several times until an equilibrium is reached; the final product, called by the Spanish Conquistadors *tumbaga*, is an object looking like gold and with the luster of gold [10,15].

In a similar manner, the phenomenon of depletion silvering may be described, in which a similar procedure is carried out on a silver object, where surface enrichment of the silver will make it appear as pure silver. It should be observed that the natural process of silver oxidation and corrosion can be easily interpreted as depletion silvering.

## 3. Materials

Forty-three nose ornaments were analyzed, of which thirty-nine contained areas of gold and silver (002 to 033 and from 102 to 112). Two of the ornaments consisted of only gold alloy (108 and 109), and two consisted of only silver alloy (005 and 110).

The 39 gold-silver nose ornaments have masses ranging from 3.5 to 9 g, with a mean value of 5.2 g. The typical size of each ornament was $7 \times 4$ cm$^2$. The thicknesses of the sheets, calculated assuming a mean density of 15 g/cm$^3$ [11], ranged from 70 to 200 μm with a mean value of 120 μm.

These nose ornaments typically represented warriors, various types of land and sea animals and hunting scenes. Among the land animals were cats, iguanas, Peruvian hairless dogs, pelicans, macaws, owls, high-altitude lizards, chinchillas and scorpions. The aquatic animals included crustaceans, and well represented are prawns. Nose ornaments 003 and 006 represent geometrical figures.

The 43 nose ornaments were part of the mummy bundle of the Lady of Cao. They were found wrapped in a cotton textile and were left over the chest of the Lady of Cao. Because of their exceptional conservation, the nose ornaments did not require any chemical treatment, and only a mechanical cleaning was necessary. Most are in exhibition in the Museo of Cao; others are stored. In both situations, they are under controlled weather conditions.

## 4. Methods

The following non-destructive and non-invasive X-ray-based (bremsstrahlung or monoenergetic) techniques were used to determine the composition of the nose decorations, to differentiate gold from gilded copper, or tumbaga, and to determine the thickness of the sheets:

**Energy Dispersive X-Ray Fluorescence (EDXRF)**. This analytical technique is described in detail elsewhere [10–15]. For this work, handheld EDXRF equipment was employed and is composed of a small dedicated X-ray tube working at 40 kV and 200 μA (see Figure 2). The X-rays are filtered with a Ba-oxide disk to produce a quasi-monoenergetic beam. The X-rays are measured with a Si-drift detector having an energy resolution of approximately 130 eV at 5.9 keV. The X-ray spectra were collected and processed on a PC. Because of the X-ray attenuation, EDXRF determines the composition of the first 5–10 μm of these objects. This allowed us to determine the homogeneousness of the alloy surface. Each nose ornament was analyzed by EDXRF at numerous different locations on both sides to highlight inhomogeneities. Each EDXRF measurement lasted 50–100 s.

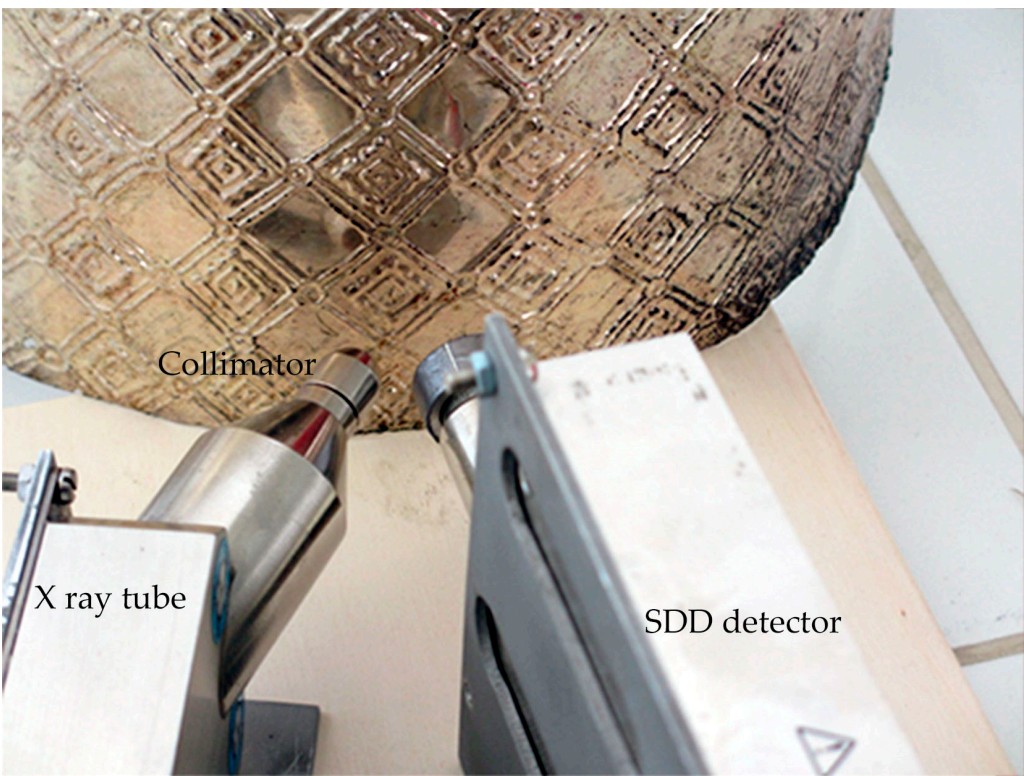

**Figure 2.** Portable EDXRF equipment for EDXRF analysis and for linear scanning, composed of a Si-drift detector (on the right; 123 SDD: 450 μm thickness, 7 mm$^2$ area and 130 eV energy resolution at 5.9 keV) and an X-ray tube (on the left; 40 kV, 100 μA maximum voltage and current and Ag-anode). Bias supply and MCA are inside the detector and of the X-ray tube, respectively. A typical measurement takes 50 s. The X-ray tube is collimated and irradiates an area of <1 mm$^2$.

The internal ratio method was used when processing the EDXRF-spectra, which is based on the following ratios [10,15]: Cu(Kα/Kβ), Au(Lα/Lβ), Au(Lα/CuKα) for the cases of depletion gilding and Cu(Kα/Kβ), Ag(Kα/Kβ), Ag(Lα/Cu Kα) for the cases of depletion silvering [10,15]. The use of these ratios, all depending on the Au or Ag-thickness, differentiates gold from gilding but gives poor results in the case of silvering, mainly because of the limited efficiency of typical Si-drift detectors for detecting the Ag L X-rays along with the background due to the Ag-K lines.

**Transmission of Monoenergetic X-Rays (XRT).** This method was used to differentiate gold or silver sheets from gilded or silvered copper and determine the sample thickness. The XRT-device is described in previous papers [18–21] and is mainly composed of an X-ray tube, working at 50 kV and 200 μA maximum voltage and current, with a secondary Sn (Sn Kα-line has an energy of 25.3 keV) and a Si-drift detector with a thickness of 1 mm and an energy resolution of 140 eV at 5.9 keV [18]. The following nose ornaments were studied by XRT 002, 003, 006, 007, 010, 011, 012, 013, 017, 018, 020, 023 and 029.

**Radiography (RAD)**. Devices for radiography are discussed in previous papers [10,11,16,19,21]. The portable device employed for studying the nose ornaments is mainly composed of an X-ray tube, working at 50 kV and 200 µA maximum voltage, current and with a flat panel detector. Radiography is able to visualize the discontinuities and inhomogeneities of a sample in terms of different attenuation of the bremsstrahlung X-rays emitted by the X-ray tube. This method was applied to the following nose ornaments: 002, 006, 011, 014, 018, 021, 022, 023, 024, 027, 028, 030, 00105 and 00106 [16,19].

## 5. Results

### 5.1. EDXRF Analysis of Gold and Silver Nose Ornaments

The results obtained from EDXRF analysis of the golden areas of the forty-one nose ornaments are listed in Table 1 and shown in Figure 3 [10–13]. An XRF spectrum of a reference sample obtained with the equipment shown in Figure 2 is shown in Figure 4. Following [12], the mean values were calculated: Au = (80.0 ± 2.5)%; Ag = (16 ± 2)%; Cu = (4 ± 1.5)%. The gold concentration would currently correspond to 19 karats (100% gold = 24 karats).

**Table 1.** Composition of gold areas of 41 nose ornaments from the tomb of the Lady of Cao.

| Number | Au (%) | Ag (%) | Cu (%) | Number | Au (%) | Ag (%) | Cu (%) |
|---|---|---|---|---|---|---|---|
| 002 | 75.5 | 19 | 5.5 | 024 | 78 | 17 | 5 |
| 003 | 79.5 | 16 | 5.5 | 025 | 80 | 16 | 4 |
| 004 | 82 | 14.5 | 3.5 | 026 | 80 | 15 | 5 |
| 006 | 79.5 | 16 | 4.5 | 027 | 79 | 16 | 5 |
| 007 | 80.5 | 15.5 | 4 | 028 | 77 | 20 | 3 |
| 008 | 82 | 15 | 3 | 029 | 82 | 12.5 | 5.5 |
| 009 | 84 | 14.5 | 1.5 | 030 | 81 | 14 | 5 |
| 010 | 82 | 14.5 | 3.5 | 031 | 77 | 17.5 | 4.5 |
| 011 | 81.5 | 14 | 4.5 | 032 | 77.5 | 17.5 | 5 |
| 012 | 80 | 16 | 4 | 033 | 81 | 17 | 2 |
| 013 | 78 | 19 | 3 | 102 | 82.5 | 13 | 4.5 |
| 014 | 81 | 14 | 5 | 103 | 80 | 14 | 6 |
| 015 | 75.5 | 20.5 | 4 | 104 | 85.5 | 11.5 | 3 |
| 016 | 79.5 | 16 | 4.5 | 105 | 80 | 14.5 | 5.5 |
| 017 | 78.5 | 16.5 | 5 | 106 | 81 | 17 | 2 |
| 018 | 75 | 19.5 | 5.5 | 107 | 72.5 | 24.5 | 3 |
| 019 | 78 | 17 | 5 | 108 | 82.5 | 12.5 | 5 |
| 020 | 82.5 | 14 | 3.5 | 109 | 80 | 13.5 | 6.5 |
| 021 | 81.5 | 14.5 | 4 | 111 | 81.5 | 14.5 | 4.5 |
| 022 | 79 | 15.5 | 5.5 | 112 | 81 | 13.5 | 5.5 |
| 023 | 82 | 14.5 | 3.5 | Mean value | 80 ± 2.5 | 16 ± 2 | 4 ± 1.5 |

The total uncertainty associated with the EDXRF experimental measurements of gold—areas containing Au, Ag and Cu—concentration depends on various reasons, such as the manufacturing method, the restoration treatment, the analytical method (EDXRF), the analysis of the EDXRF-spectrum and the reference samples employed for EDXRF analysis. All these contributions are discussed in detail in par. 3.1.5 of Ref. [10].

The total uncertainty associated with the EDXRF experimental measurements of silver—areas containing Ag, Cu and Au—concentration is much more critical, mainly depending on the surface enrichment processes at the surface.

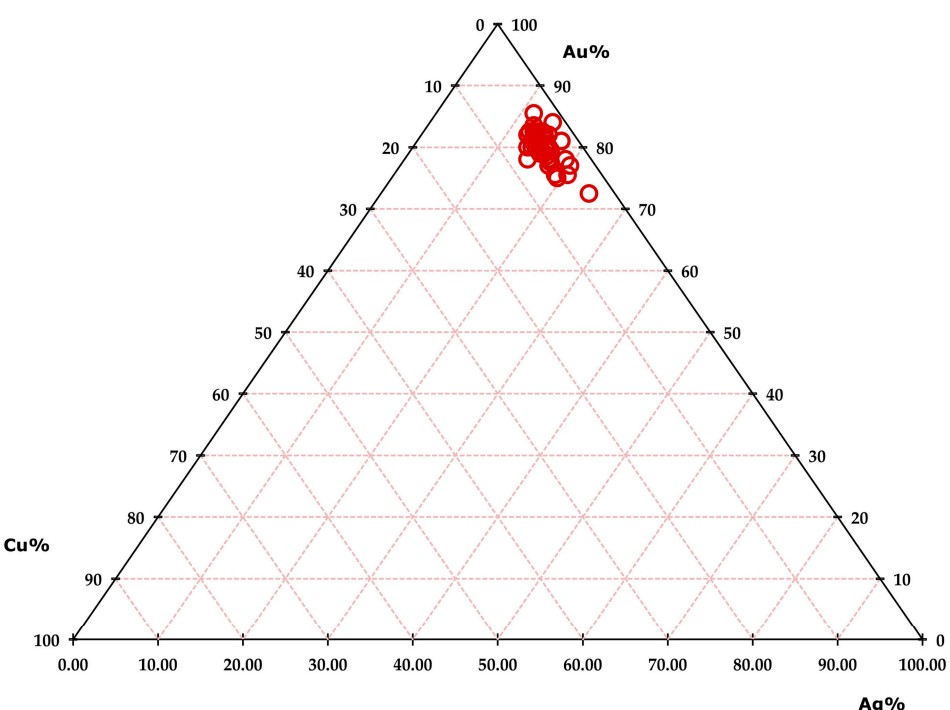

**Figure 3.** Distribution of EDXRF measurements of the 41 golden areas of the 43 nose ornaments from the tomb of the Lady of Cao.

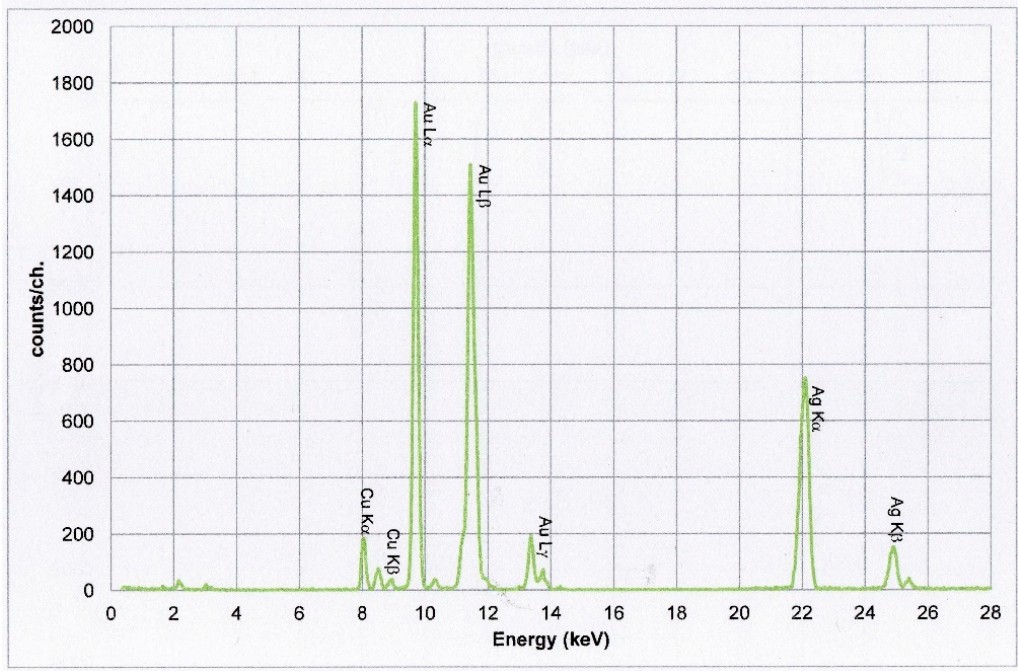

**Figure 4.** Typical X-ray spectrum of an Au-Ag-Cu alloy reference sample (70% Au, 25% Ag, 5% Cu) obtained with the equipment shown in Figure 2. The X-ray peaks are, from left: Cu-Kα line (8 keV), Au-Ll (8.5 keV), Cu-Kβ (8.9 keV), Au-Lα (9.7 kev), Au-Lβ (11.5 keV), Au-Lγ (13.4 and 13.8 keV), Ag-Kα (22 keV) and Ag-Kβ (25 keV). The small peaks between Cu-Kα and Cu-Kβ and between Au-Lα and Au-Lβ correspond to Au-Lη (8.5 keV) and Au-Ll (10.3 keV) lines, which have an intensity of 5% and 2.5%, respectively, considering 100 as the intensity of Au-Lα line.

It is certainly true that the jewels examined present significant differences in production technology, especially as regards the techniques of joining the sheets. Since technological

innovations take time, jewels were not all produced at the same time. Some of them were certainly created for the burial of the Lady of Cao because they have very similar compositions and manufacture. This shows that the Moche probably had good control of the alloying processes in order to obtain a certain composition over time.

The situation regarding the silver areas is quite different. Those areas were mainly silver, with copper and gold at lower levels. The composition of the 41 silver areas of the nose ornaments is shown in Figure 5, showing their large silver content variation [10,11].

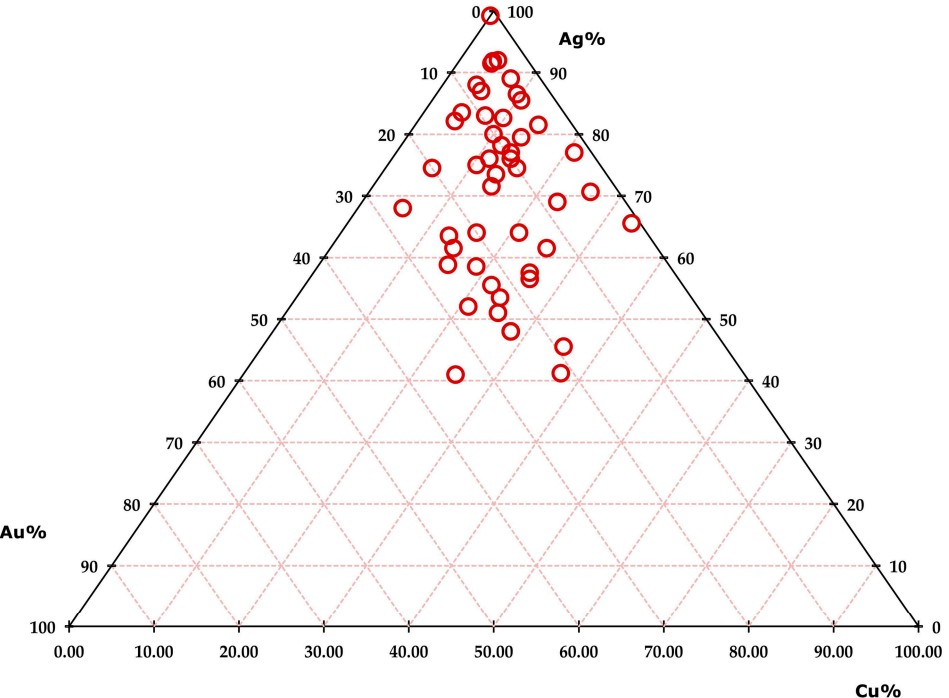

**Figure 5.** Distribution of EDXRF measurements of the 41 silver areas of the 43 nose ornaments, from 2 to 112. The large spread of concentration and the high gold concentration are remarkable.

*5.2. Attenuation Measurements Using Monoenergetic X-rays*

From attenuation measurements, the thickness of the gold areas was measured as being approximately 100 µm for five nose ornaments (002, 011, 013, 017 and 023). In seven cases (003, 006, 008, 010, 012, 020 and 029), the attenuation of photons through the gold sheet was almost complete, indicating a gold thickness d > 100 µm.

Further, also from attenuation measurements, the mean thickness of the silver areas of nose ornaments 002, 003, 006, 008, 010, 011, 012, 013, 017, 020, 023 and 029 was measured as being 200 ± 75 µm.

It is extremely unlikely that the golden areas have all been subjected to a depletion gilding process, giving a similar composition. Furthermore, a gilding process was not detected from an altered Au-Lα/Au-Lβ ratio [15]. Finally, the radiographs of many nose ornaments show that the gold areas are relatively homogeneous and that there is a clear visual attenuation difference between gold and silver areas.

In principle, the possibility of depletion silvering of some nose ornaments cannot be excluded. However, in this case, the internal ratio method is hardly applicable [15]. The radiographs of many nose ornaments show that the silver areas are intrinsically inhomogeneous, but the difference between gold and silver areas is very clear, both visually and in terms of grey level [12].

**6. Discussion**

By considering the results shown in Table 1, demonstrating a similar composition of gold areas of all 41 nose ornaments, it is extremely unlikely that the golden areas have

all been subjected to a depletion gilding process, giving rise to similar final compositions. Further, there is no evidence of a gilding process from an altered (Au-Lα/Au-Lβ)-ratio, which depends on the Au-thickness [15]. Finally, the radiographs of many nose ornaments show that the golden areas are relatively homogeneous and that there is a clear visual difference between the gold and silver areas.

On the other hand, the composition of the 41 silver areas of the nose ornaments, shown in Figure 5, demonstrates a large spread in the concentration of the metals [10,11]. For these areas, the possibility of the depletion silvering technique having been used cannot be excluded. Therefore, the internal ratio method is hardly applicable [10]. However, the radiographs of many nose ornaments show that the silver areas are intrinsically inhomogeneous, but the difference between gold and silver areas is large in [11] and contradict the hypothesis of Ingo et al. [17].

Furthermore, according to the hypothesis of Ingo et al., a **unique sheet** of a gold-silver-copper alloy [17] was treated with a separate depletion gilding and silvering process. This process would remove copper and silver from the surface. By removing just copper, layers containing gold and silver should be produced. Depletion (gilding or silvering) generally removes copper and/or silver from the inner surface of the alloy [17], leaving the bulk unchanged. Therefore, the attenuation of the alloy sheets of the nose ornament should be approximately the same. If the depletion extends beyond the surface, the attenuation of X-rays on the silver side must be lower than that of the gilding side. Further, if depletion moves all requested metals, copper in the case of silvering and copper and silver in the case of gilding, the attenuation of the silver side should be lower than that of the gold side. The radiographs showed a significant attenuation difference between the silver and gold areas, with the gold side having more attenuation.

The transmission measurements (see Refs. [15,20]) resulted in significant differences in the attenuation of gold and silver areas, as was expected if these areas were actually made on gold and silver alloys. As hypothesized by Ingo of gold subject to depletion gilding, both Ag and Cu present in the Au-alloy would emigrate to the surface, being removed. In the Ag-areas, only Cu would emigrate to the surface, being removed. Finally, the attenuation of gold and silver areas would be similar, starting from the same layer. This is not confirmed from the transmission experiments, also given that the thickness of the gold and silver alloys is comparable.

Finally, the hypothesis of the ornaments being created from a unique, three-component sheet of Au-Ag-Cu alloy, with the different areas treated with the surface techniques of either silver depletion gilding [22–24] or silvering, should not result in a gold-silver interface of the bulk material before or after depletion. However, all studies carried out on the Au-Ag interfaces, both by transmission measurements and by radiographs, show a sharp discontinuity (see Figures 6–12). All measurements carried out on Au-Ag interfaces by transmission of monoenergetic X-rays show a similar behavior. Unlike the hypothesis of one sheet and no interface, four different processes of joining gold and silver sheets were identified by using EDXRF, XRT and RAD:

1.  Gluing or mechanical joining, as in the case of nose ornament 002 (see Figure 6a showing two Ag-buttons for the two shields). The silver sheets are joined to the gold body with a "button" (visible in Figure 6), while the hats are probably glued.
2.  Soldering, using a silver alloy (sometimes a silver-copper alloy), as in the case of nose ornament 105 (see Figure 7). The soldering with Ag is demonstrated by the EDXRF-spectrum of the soldering area, showing an excess of Ag.
3.  Using a mercury amalgam with a heating process, as in the case of nose ornament 011 [25], where Hg was clearly detected in the interface of Au-Ag and possibly in many other nose ornaments.
4.  Using a hammering-heating process with a temperature sufficient to soften the silver alloy, as in the case of nose ornament 024.

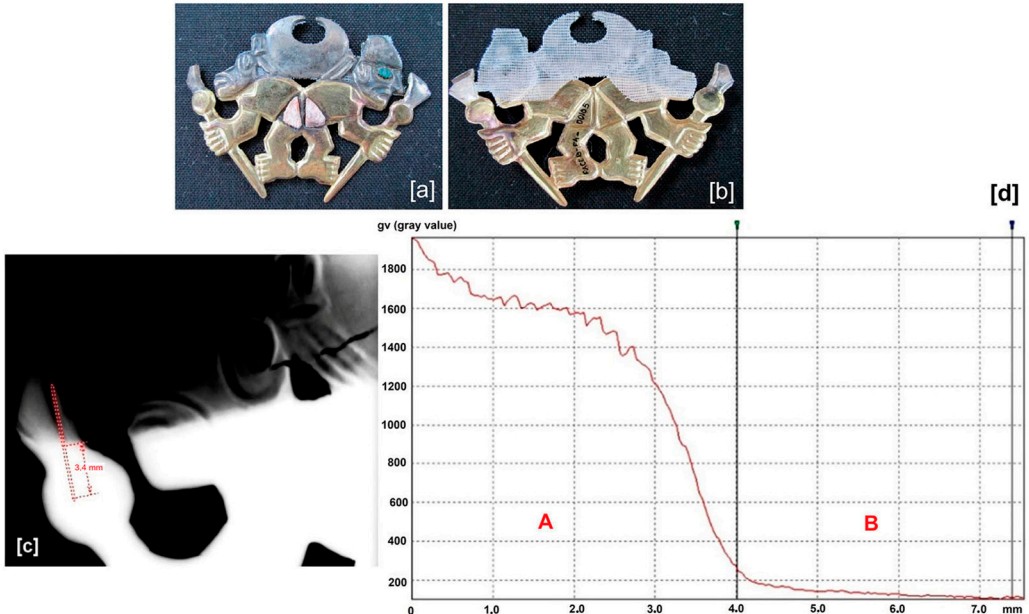

**Figure 6.** Images (**a**–**c**) show nose ornament 002, while image (**d**) shows the related radiography along with gray level profile along red line. The two silver hats were possibly glued to the golden body, while the two silver shields were possibly pressed to the body like a button. The last Figure shows, from the left, the level of the silver hat alone (A), the level of the hat superimposed onto the golden body (B).

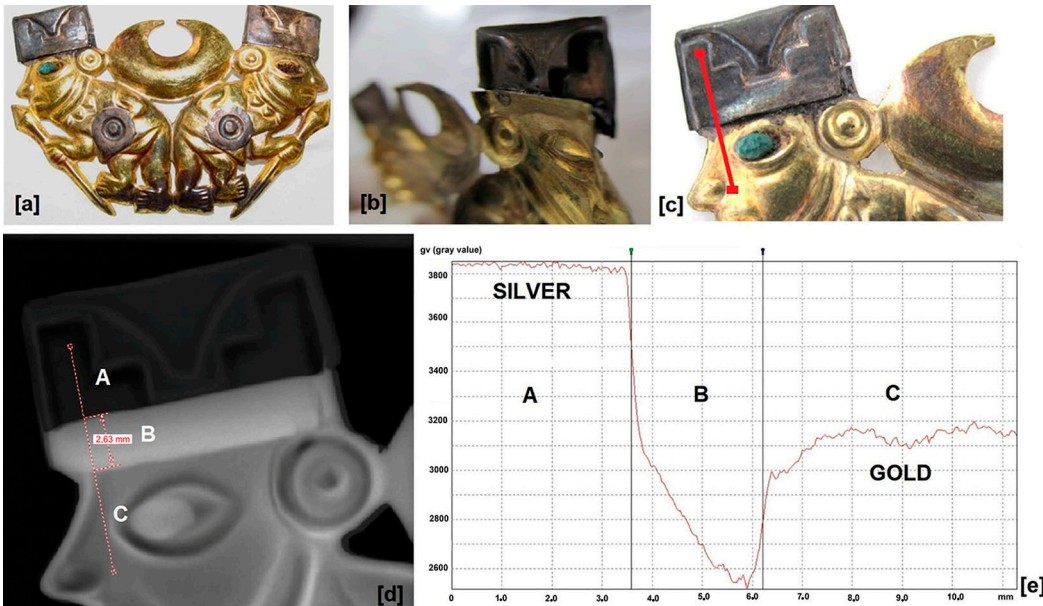

**Figure 7.** Front and rear side of nose ornament 105 (**a**,**b**), its radiography and the grey level profile (along red line) of the gold-silver interface (**c**,**d**). A clear difference may be observed between the gold and the silver areas. This nose ornament is in bad condition and was undergoing restoration. The fragment studied by Ingo et al. was taken from the face on the top right of Figure (**a**). The soldering material was an Ag-Cu alloy, as shown by EDXRF-measurement (**e**).

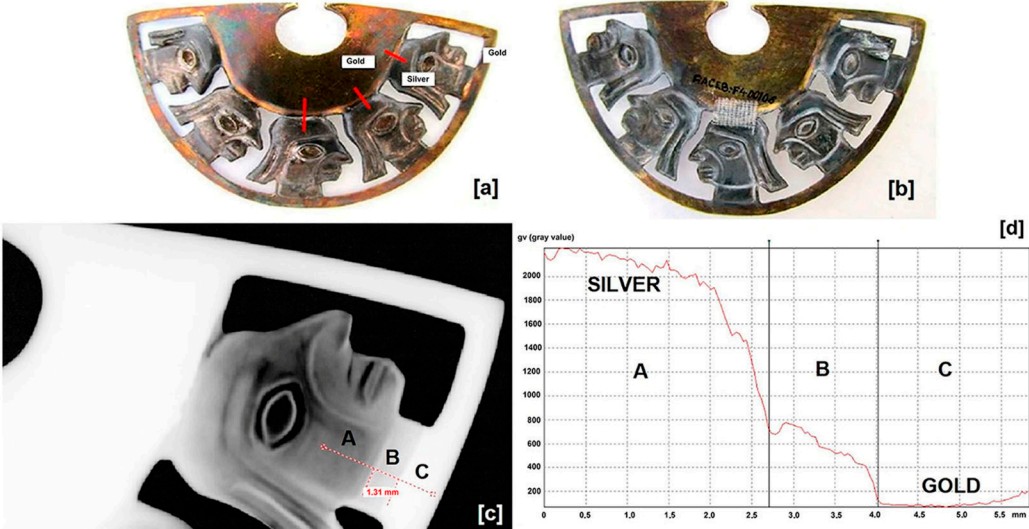

**Figure 8.** Nose ornament 106 (top figures, front and rear) (**a**,**b**) and its radiography with grey numbers (**c**). The radiography clearly shows that the five silver heads have been soldered to the gold semi-circles on both sides. The scan along the red line (grey levels) shows from the right the golden area, the soldering area (B) and the silver area (A) (**d**).

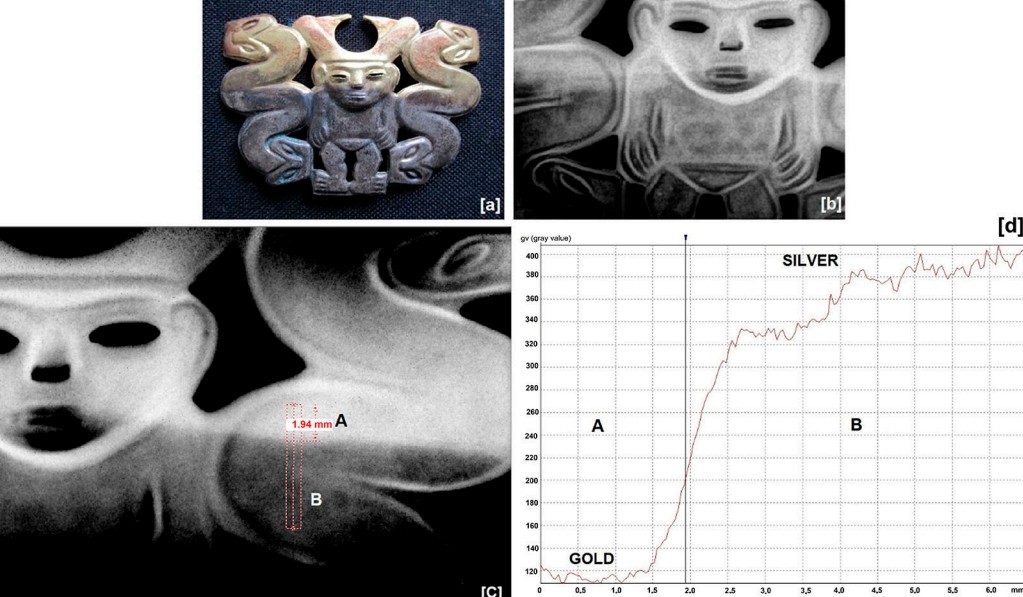

**Figure 9.** Images (**a**–**c**) representing the front side of nose ornament 022 and the related radiography (**d**) with the corresponding grey level profile along the red line shows the clear discontinuity between grey level profile along the red line shows the clear discontinuity between the gold and silver areas.

The described methods to join gold and silver sheets have been reproduced experimentally. The behavior of the Au-Ag interface was always the same as typically described by the red curves in Figures 9, 11 and 12.

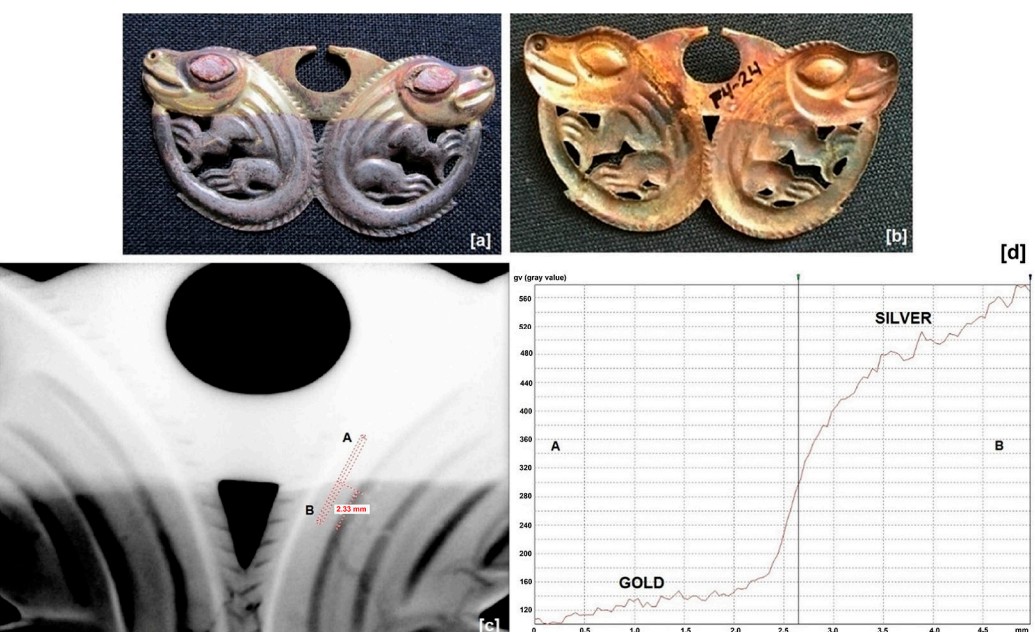

**Figure 10.** Images (**a**–**c**) representing the front side of nose ornament 024 and the related radiog-raphy (**d**) with the corresponding grey level profile along the red line (A-B in Figure 7c). This shows the clear discontinuity between the gold and silver areas.

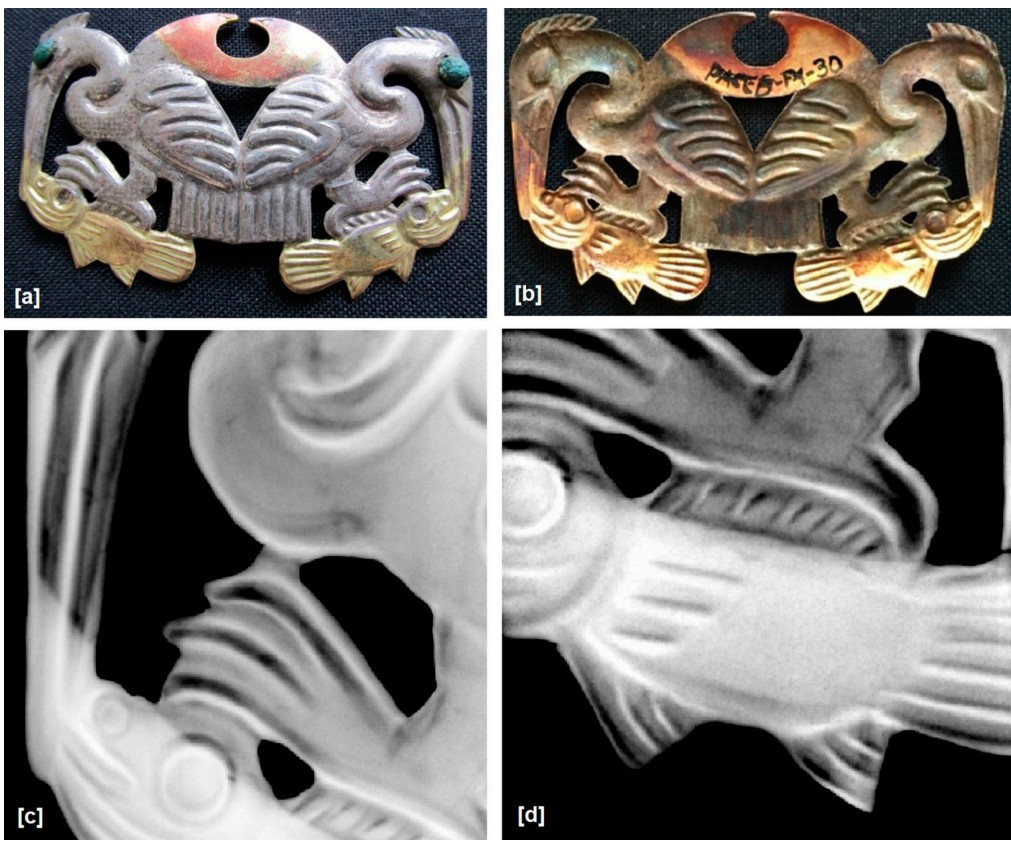

**Figure 11.** Images (**a**,**b**) representing the front and back side of nose ornament 030 and the related radiography (**c**,**d**) showing the discontinuity between gold and silver areas. It may be that the gold-silver junction has been produced by a Hg amalgam.

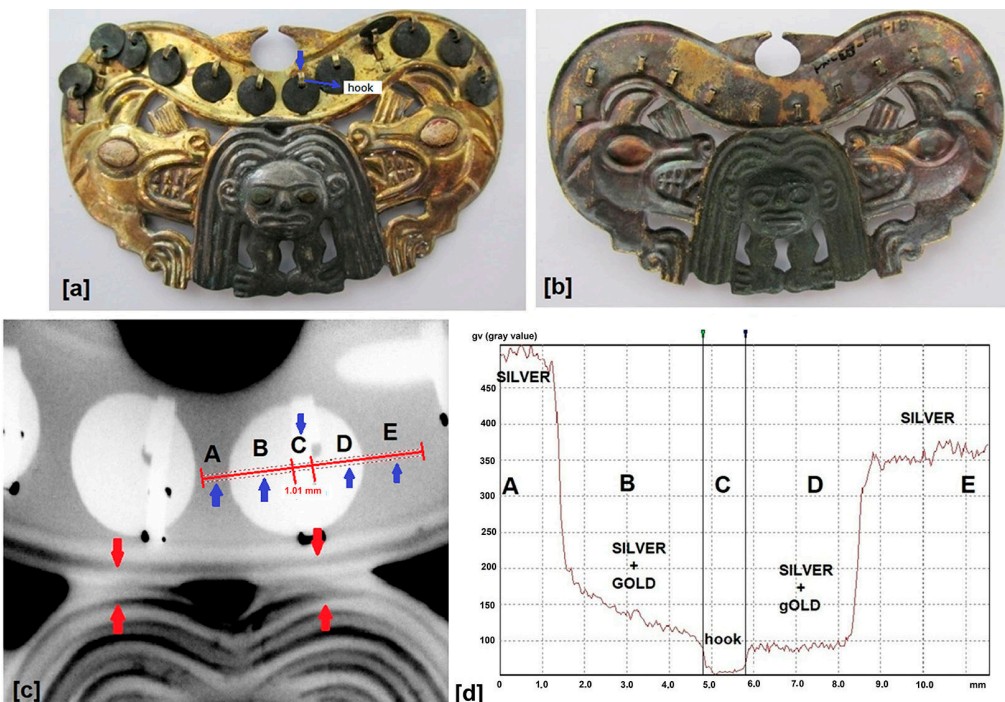

**Figure 12.** (**a**) The front and (**b**) rear sides of nose ornament 018 are perplexing. The gold areas on the rear side are partially blackened. (**c**) The radiography shows an unclear discontinuity between gold and silver (hair) and (**d**) the grey level of the red line interval A–E.

In this context, the 39 Au-Ag nose decorations were revised and partially subdivided by us, also taking into account their external aspect, results of EDXRF analysis and the other employed methods.

A different case is ornaments 002 and 006, where silver and gold sheets are glued together or mechanically joined. In 002, two shields and two hats on almost pure silver have been mechanically joined or glued to the golden body (see Figure 6). Finally, there are the nose ornaments in which gold and silver sheets are soldered together: 008, 010, 012, 017, 032 and 106. As an example of the last case, five silver heads are clearly soldered to the gold alloy body (see Figure 8). The soldering material was an Ag-Cu alloy, as shown by the EDXRF measurement.

In another case, similar nose ornaments, 004, 021 and 107, are composed of a gold body with a silver decoration strip on the bottom.

The following nose ornaments appear to be of more recent production and, therefore, of possibly more recent production, where the two sheets of Au and Ag are not glued or soldered: 007, 009, 011, 013, 016, 018,019, 022 to 031, 033 and 105. The radiographs of nose ornaments 022, 024 and 030 are shown in Figures 9–11. A Hg amalgam was clearly detected only in nose decoration 011, but its absence is not a demonstration that an Hg amalgam was not employed. We suspect that, for nose ornaments of this group, the junction was actually produced by using a Hg amalgam. Figure 11 shows, for example, the discontinuity of gold-silver in nose decoration 030.

Finally, nose ornament 018 is anomalous, which possibly refers to an object subjected to an unsuccessful depletion gilding process, resulting in a highly corroded state. There are clearly visible traces of gilding on the rear side, and the radiography does not show a significant difference between gold and silver areas, as usually found (Figure 12). This is a strange and difficult case and all our studies conflict with each other. Analysis of this ornament's gold areas by EDXRF gives the same composition as all other nose ornaments, while silver areas show a relatively low concentration of Au and Cu. The silver disks have the same composition as the silver body. Finally, both $Cu(K\alpha/K\beta)$ and $Au(L\alpha/L\beta)$ values are not compatible with a gilded object.

### 7. Conclusions

Forty-three nose ornaments from the Moche tomb of the Lady of Cao have been studied using various methods. Of these 43 ornaments, 39 were manufactured by joining sheets of gold to sheets of silver alloys, 2 were on silver alloys and 2 on gold alloys. The following analytical steps were ascertained:

A.  All gold areas have a similar composition.
B.  Silver areas have quite an erratic composition.
C.  Silver areas show, besides silver and copper, an unusually high concentration of gold of up to 35%.

The conclusion using the prevailing technology in the production of the artifacts is that of joining together two sheets, one of gold alloy and the other of silver. Therefore, the techniques they use is not that of a systematic depletion on a single sheet, a technique which could have been used in some cases, for the following main reasons:

*   Radiographs, carried out on about 15 nose ornaments, clearly show that two separate sheets of gold and silver alloy had to be used and joined together.
*   Attenuation measurements with monoenergetic radiation, which quantitatively confirmed not only the hypothesis of two sheets of Au and Ag, but also determined that the two sheets have different thicknesses (Ag-sheets are approximately two times thicker than the Au-sheets).

EDXRF analysis, which showed that the internal ratios (typically Au-L$\alpha$/AuL$\beta$ or Ag-K/Cu-K) are generally compatible with the hypothesis of two separate sheets of Au and Ag and not with that of a unique sheet subject to depletion gilding or silvering. However, it cannot be completely excluded that one of the few artifacts specifically studied by Ingo and co-workers, i.e., 00105 (but suspect it is also nose ornament 018), could have been realized starting from a unique sheet of Cu-Au-Ag alloy, subject to depletion gilding (golden areas) or silvering (silver areas).

Therefore, on the basis of the following techniques, EDXRF, XRT and RAD, it can be concluded that the thirty-seven nose ornaments from the tomb of the Lady of Cao could have been created starting from a unique sheet of Cu-Au-Ag alloy.

**Author Contributions:** Conceptualization, R.C. and R.F.; Methodology, G.E.G. and R.C.; Software, S.A. and R.T.L.; Validation, A.B., A.F., S.A. and R.T.L.; Formal analysis, S.A.; Investigation, G.E.G., A.B., A.F., S.A. and R.T.L.; Resources, A.F. and R.F.; Data curation, G.E.G. and A.B.; Writing—original draft, R.C.; Writing—review & editing, G.E.G.; Visualization, A.F.; Supervision, R.C.; Project administration, R.C. and A.B. All authors have read and agreed to the published version of the manuscript.

**Funding:** This research received no external funding.

**Data Availability Statement:** Not applicable.

**Acknowledgments:** The authors thank Albert Hanson for careful revision of the text.

**Conflicts of Interest:** The authors declare no conflict of interest.

## Appendix A

**Table A1.** The forty-three nose ornaments from the Moche tomb of the Lady of Cao.

| Object picture (front and, in a few cases, back side) | Code (PACEB-F4-00X) and<br>Gold composition: Au(%), Ag(%), Cu(%)<br>Silver composition: Ag(%), Cu(%), Au(%) |
|---|---|
| 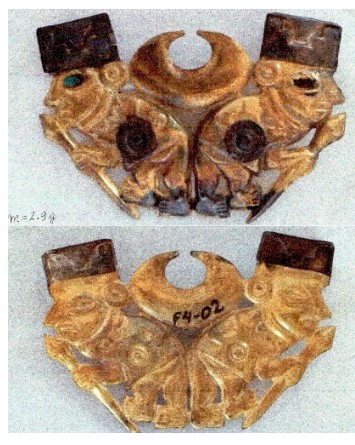 | PACEB-F4-002 (gold and silver alloy)<br>Gold: 75.5, 19, 5.5<br>Silver: 99, 0.2, 0.8 |
| 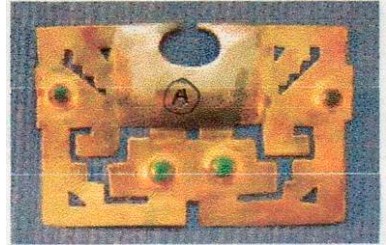 | PACEB-F4-003 (gold and silver alloy)<br>Gold: 79.5, 16, 5.5<br>Silver: 83, 7.5, 9.5 |
| 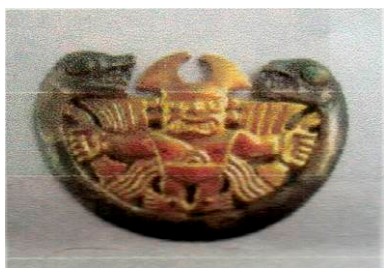 | PACEB-F4-004 (gold and silver alloy)<br>Gold: 82, 14.5, 3.5<br>Silver: 92, 4.5, 3.5 |
| 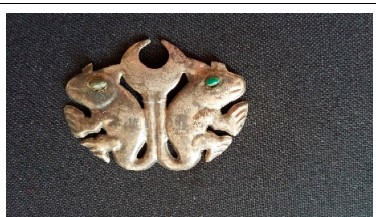 | PACEB-F4-005 (silver alloy)<br>Silver: 70, 20, 10 |

Museum LADY OF CAO (Magdalena de Cao, Trujillo, Peru)
The site Museum Lady of Cao gives a complete overview of the El Brujo Archaeological Complex and the many ancient objects that were found there, including funeral objects, tools and gold and silver jewelry. The tomb of the Lady of Cao was discovered in 2006 by a team of Peruvian archaeologists directed by Régulo Franco Jordán. The 1500-year-old mummy of the Lady of Cao, heavily tattooed and well preserved, shed new light on the Moche culture, which occupied Peru's northern coast from about 100 to 700 CE.

**Table A1.** *Cont.*

Museum LADY OF CAO (Magdalena de Cao, Trujillo, Peru)
The site Museum Lady of Cao gives a complete overview of the El Brujo Archaeological Complex and the many ancient objects that were found there, including funeral objects, tools and gold and silver jewelry. The tomb of the Lady of Cao was discovered in 2006 by a team of Peruvian archaeologists directed by Régulo Franco Jordán. The 1500-year-old mummy of the Lady of Cao, heavily tattooed and well preserved, shed new light on the Moche culture, which occupied Peru's northern coast from about 100 to 700 CE.

| Object picture (front and, in a few cases, back side) | Code (PACEB-F4-00X) and Gold composition: Au(%), Ag(%), Cu(%) Silver composition: Ag(%), Cu(%), Au(%) |
|---|---|
| 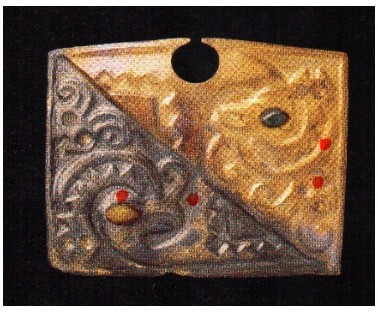 | PACEB-F4-006 (gold and silver alloy) Gold: 79.5, 16, 4.5 Silver: 92, 4, 4 |
| 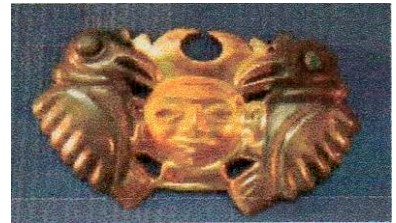 | PACEB-F4-007 (gold and silver alloy) Gold: 80.5, 15.5, 4 Silver: 87, 5, 8 |
| 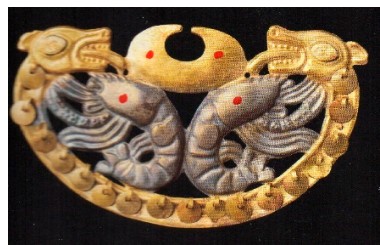 | PACEB-F4-008 (gold and silver alloy) Gold: 82, 15, 3 Silver *: 50, 20, 30 |
| 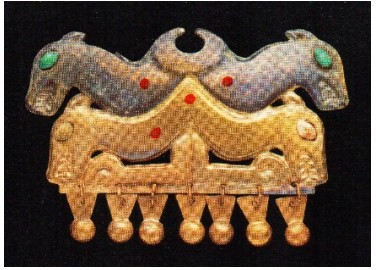 | PACEB-F4-009 (gold and silver alloy) Gold: 84, 14.5, 1.5 Silver *: 80, 12, 8 |
| 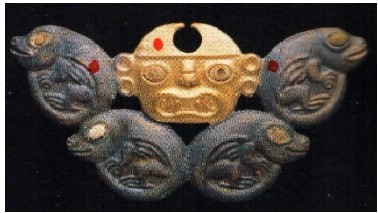 | PACEB-F4-010 (gold and silver alloy) Gold: 82, 14.5, 3.5 Silver *: 60, 15, 25 |

**Table A1.** *Cont.*

| Museum LADY OF CAO (Magdalena de Cao, Trujillo, Peru) |
|---|
| The site Museum Lady of Cao gives a complete overview of the El Brujo Archaeological Complex and the many ancient objects that were found there, including funeral objects, tools and gold and silver jewelry. The tomb of the Lady of Cao was discovered in 2006 by a team of Peruvian archaeologists directed by Régulo Franco Jordán. The 1500-year-old mummy of the Lady of Cao, heavily tattooed and well preserved, shed new light on the Moche culture, which occupied Peru's northern coast from about 100 to 700 CE. |

| Object picture (front and, in a few cases, back side) | Code (PACEB-F4-00X) and Gold composition: Au(%), Ag(%), Cu(%) Silver composition: Ag(%), Cu(%), Au(%) |
|---|---|
| 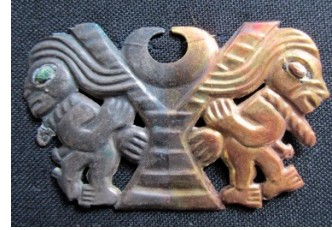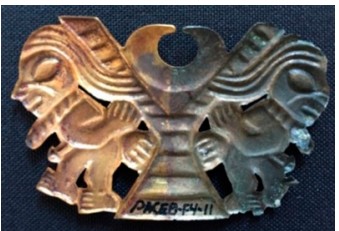 | PACEB-F4-011 (gold and silver alloy) Gold: 81.5, 14, 4.5 Silver: 62, 23, 15 |
| 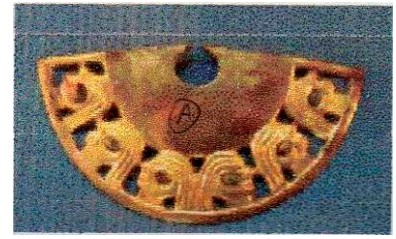 | PACEB-F4-012 (gold and silver alloy) Gold: 80, 16, 4 Silver: 46, 35, 19 |
| 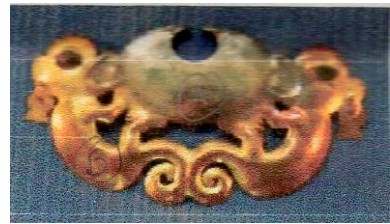 | PACEB-F4-013 (gold and silver alloy) Gold: 78, 19, 3 Silver: 85, 11, 4 |
| 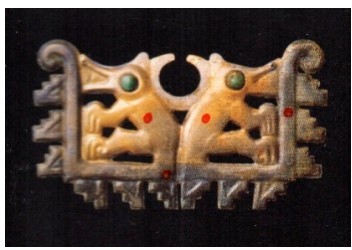 | PACEB-F4-014 (gold and silver alloy) Gold: 81, 14, 5 Silver: 60, 19, 21 |
| 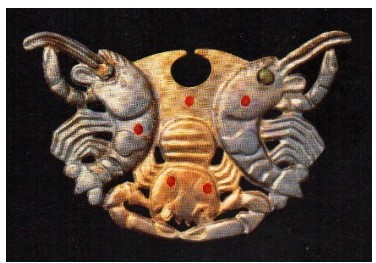 | PACEB-F4-015 (gold and silver alloy) Gold: 75.5, 20.5, 4 Silver: 62, 14, 24 |

**Table A1.** *Cont.*

Museum LADY OF CAO (Magdalena de Cao, Trujillo, Peru)
The site Museum Lady of Cao gives a complete overview of the El Brujo Archaeological Complex and the many ancient objects that were found there, including funeral objects, tools and gold and silver jewelry. The tomb of the Lady of Cao was discovered in 2006 by a team of Peruvian archaeologists directed by Régulo Franco Jordán. The 1500-year-old mummy of the Lady of Cao, heavily tattooed and well preserved, shed new light on the Moche culture, which occupied Peru's northern coast from about 100 to 700 CE.

| Object picture (front and, in a few cases, back side) | Code (PACEB-F4-00X) and<br>Gold composition: Au(%), Ag(%), Cu(%)<br>Silver composition: Ag(%), Cu(%), Au(%) |
|---|---|
| 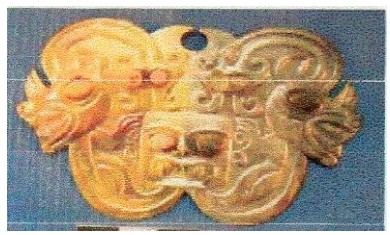 | PACEB-F4-016 (gold and silver alloy)<br>Gold: 79.5, 16, 4.5<br>Silver: 73, 14, 13<br>Hg in the Au-Ag interface |
| 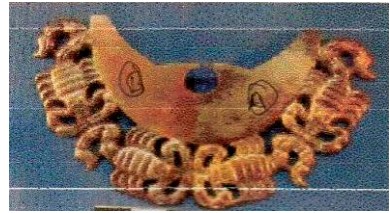 | PACEB-F4-017 (gold and silver alloy)<br>Gold: 78.5, 16.5, 5<br>Silver: 64, 16, 20 |
| 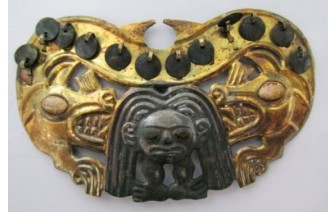 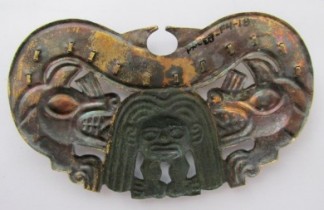 | PACEB-F4-018 (gold and silver alloy)<br>Gold: 75, 19.5, 5.5<br>Silver: 73, 19, 8; disks: 85, 8, 7 |
| 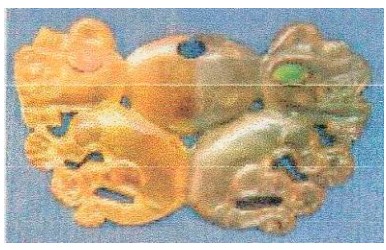 | PACEB-F4-019 (gold and silver alloy)<br>Gold: 78, 17, 5<br>Silver: 88, 4, 8 |
| 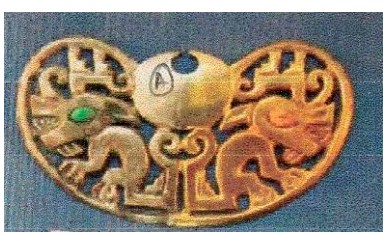 | PACEB-F4-020 (gold and silver)<br>Gold: 82.5, 14, 3.5<br>Silver: 77, 21, 2 |

**Table A1.** *Cont.*

Museum LADY OF CAO (Magdalena de Cao, Trujillo, Peru)
The site Museum Lady of Cao gives a complete overview of the El Brujo Archaeological Complex and the many ancient objects that were found there, including funeral objects, tools and gold and silver jewelry. The tomb of the Lady of Cao was discovered in 2006 by a team of Peruvian archaeologists directed by Régulo Franco Jordán. The 1500-year-old mummy of the Lady of Cao, heavily tattooed and well preserved, shed new light on the Moche culture, which occupied Peru's northern coast from about 100 to 700 CE.

| Object picture (front and, in a few cases, back side) | Code (PACEB-F4-00X) and Gold composition: Au(%), Ag(%), Cu(%) Silver composition: Ag(%), Cu(%), Au(%) |
| --- | --- |
| 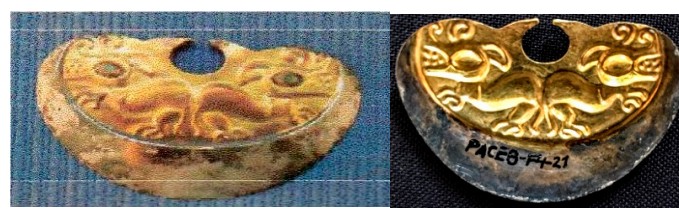 | PACEB-F4-021 (gold and silver alloy) Gold: 81.5, 14.5, 4 Silver *: 55, 30, 15 |
| 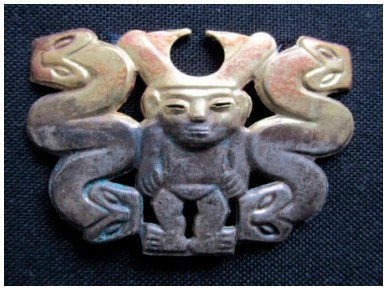 | PACEB-F4-022 (gold and silver alloy) Gold: 79, 15.5, 5.5, Silver: 68, 18, 14 |
| 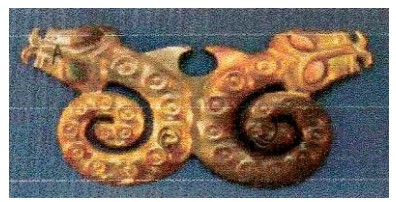 | PACEB-F4-023 (gold and silver alloy) Gold: 82, 14.5, 3.5 Silver: 76, 12, 12 |
| 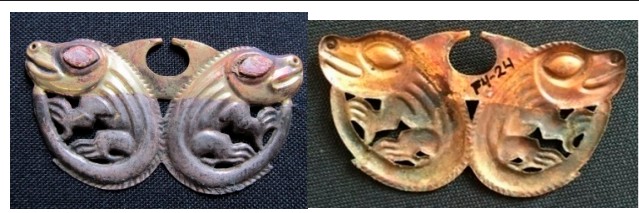 | PACEB-F4-024 (gold and silver alloy) Gold: 78, 17, 5 Silver: 56, 24, 20 |
| 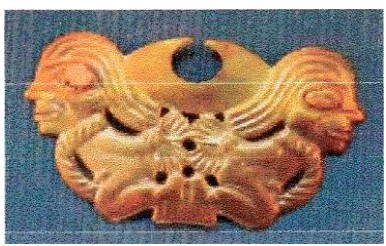 | ACEB-F4-025 (gold and silver alloy) Gold: 80, 16, 4 Silver: 56, 22, 22 |
| 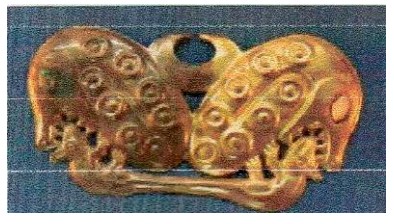 | PACEB-F4-026 (gold and silver alloy) Gold: 80, 15, 5 Silver: 77, 13, 10 |

**Table A1.** *Cont.*

Museum LADY OF CAO (Magdalena de Cao, Trujillo, Peru)
The site Museum Lady of Cao gives a complete overview of the El Brujo Archaeological Complex and the many ancient objects that were found there, including funeral objects, tools and gold and silver jewelry. The tomb of the Lady of Cao was discovered in 2006 by a team of Peruvian archaeologists directed by Régulo Franco Jordán. The 1500-year-old mummy of the Lady of Cao, heavily tattooed and well preserved, shed new light on the Moche culture, which occupied Peru's northern coast from about 100 to 700 CE.

| Object picture (front and, in a few cases, back side) | Code (PACEB-F4-00X) and Gold composition: Au(%), Ag(%), Cu(%) Silver composition: Ag(%), Cu(%), Au(%) |
|---|---|
| 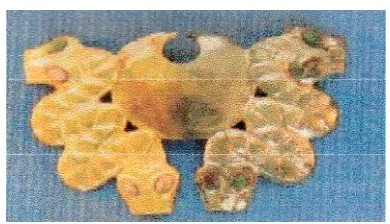 | PACEB-F4-027 (gold and silver alloy) Gold: 79, 16, 5 Silver: 82, 14, 4 |
| 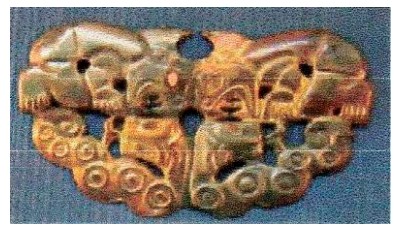 | PACEB-F4-028 (gold and silver alloy) Gold: 77, 20, 3 Silver: 89, 7, 4 |
| 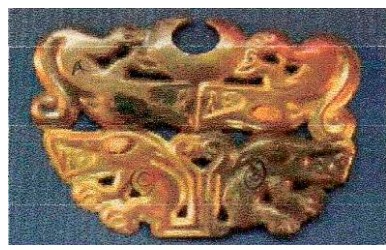 | PACEB-F4-029 (gold and silver alloy) Gold: 82, 12.5, 5.5 Silver: 57, 26, 17 |
| 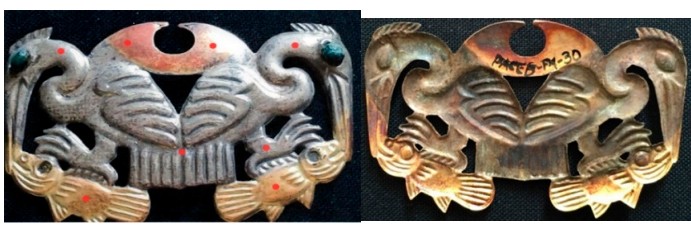 | PACEB-F4-030 (gold and silver alloy) Gold: 81, 14, 5 Silver: 52, 26, 22 |
| 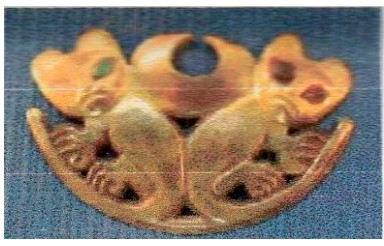 | PACEB-F4-031 (gold and silver alloy) Gold: 77, 17.5, 4.5 Silver *: 68, 16, 16 |

**Table A1.** *Cont.*

Museum LADY OF CAO (Magdalena de Cao, Trujillo, Peru)
The site Museum Lady of Cao gives a complete overview of the El Brujo Archaeological Complex and the many ancient objects that were found there, including funeral objects, tools and gold and silver jewelry. The tomb of the Lady of Cao was discovered in 2006 by a team of Peruvian archaeologists directed by Régulo Franco Jordán. The 1500-year-old mummy of the Lady of Cao, heavily tattooed and well preserved, shed new light on the Moche culture, which occupied Peru's northern coast from about 100 to 700 CE.

| Object picture (front and, in a few cases, back side) | Code (PACEB-F4-00X) and Gold composition: Au(%), Ag(%), Cu(%) Silver composition: Ag(%), Cu(%), Au(%) |
|---|---|
| 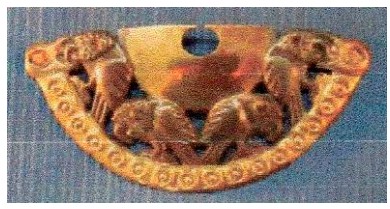 | PACEB-F4-032 (gold and silver alloy) Gold: 77.5, 17.5, 5 Silver *: 60, 19, 21 |
| 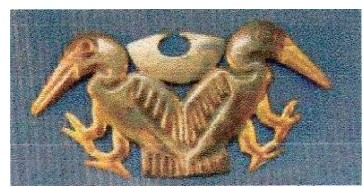 | PACEB-F4-033 (gold and silver alloy) Gold: 81, 17, 2 Silver: 76, 18, 6 |
| 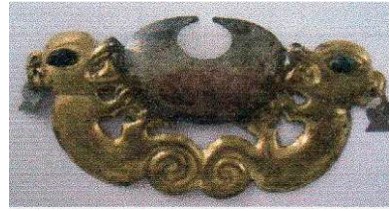 | PACEB-F4-00102 (gold and silver alloy) Gold: 82.5, 13, 4.5 Silver: 82, 4, 14 |
| 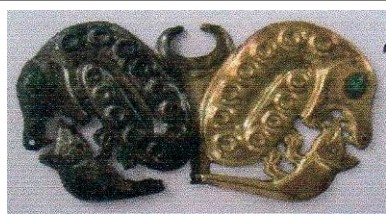 | PACEB-F4-00103 (gold and silver alloy) Gold: 80, 14, 6 Silver: 78, 12, 10 |
| 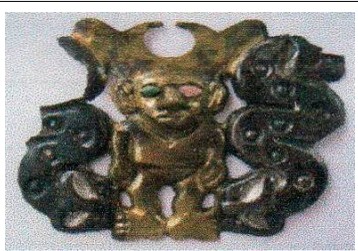 | PACEB-F4-00104 (gold and silver alloy) Gold: 85.5, 11.5, 3 Silver: 83, 10, 7 |
| 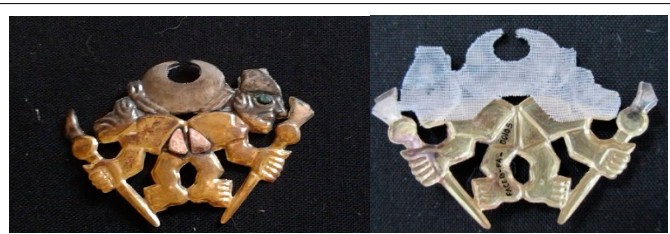 | PACEB-F4-00105 (gold and silver alloy) Gold: 80, 14.5, 5.5 Silver: 60, 15, 25 |

**Table A1.** *Cont.*

| Museum LADY OF CAO (Magdalena de Cao, Trujillo, Peru) |
| --- |
| The site Museum Lady of Cao gives a complete overview of the El Brujo Archaeological Complex and the many ancient objects that were found there, including funeral objects, tools and gold and silver jewelry. The tomb of the Lady of Cao was discovered in 2006 by a team of Peruvian archaeologists directed by Régulo Franco Jordán. The 1500-year-old mummy of the Lady of Cao, heavily tattooed and well preserved, shed new light on the Moche culture, which occupied Peru's northern coast from about 100 to 700 CE. |

| Object picture (front and, in a few cases, back side) | Code (PACEB-F4-00X) and Gold composition: Au(%), Ag(%), Cu(%) Silver composition: Ag(%), Cu(%), Au(%) |
| --- | --- |
| 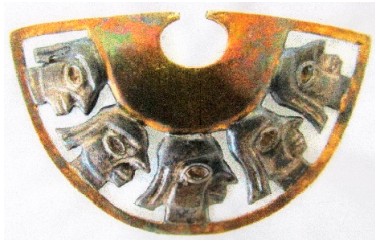 | PACEB-F4-00106 (gold and silver alloy) Gold: 81, 17, 2 Silver: 92, 4, 4 |
| 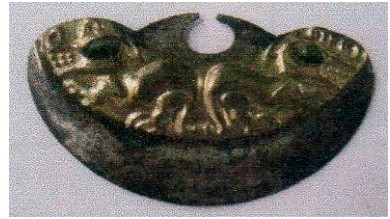 | PACEB-F4-00107 (gold and silver alloy) Gold: 72.5, 24.5, 3 Silver *: 74, 6, 20 |
| 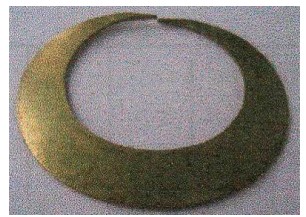 | PACEB-F4-00108 (gold alloy) Gold: 82.5, 12.5, 5 |
| 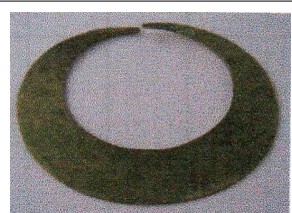 | PACEB-F4-00109 (gold alloy) Gold: 80, 13.5, 6.5 |
| 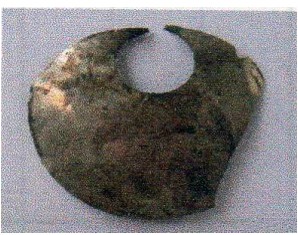 | PACEB-F4-00110 (silver alloy) Silver *: 41, 37, 22 |

**Table A1.** *Cont.*

Museum LADY OF CAO (Magdalena de Cao, Trujillo, Peru)

The site Museum Lady of Cao gives a complete overview of the El Brujo Archaeological Complex and the many ancient objects that were found there, including funeral objects, tools and gold and silver jewelry. The tomb of the Lady of Cao was discovered in 2006 by a team of Peruvian archaeologists directed by Régulo Franco Jordán. The 1500-year-old mummy of the Lady of Cao, heavily tattooed and well preserved, shed new light on the Moche culture, which occupied Peru's northern coast from about 100 to 700 CE.

| Object picture (front and, in a few cases, back side) | Code (PACEB-F4-00X) and Gold composition: Au(%), Ag(%), Cu(%) Silver composition: Ag(%), Cu(%), Au(%) |
|---|---|
| 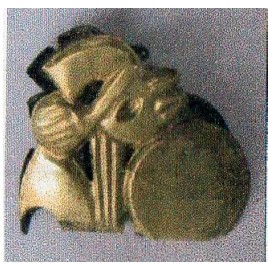 | PACEB-F4-00111 (gold and silver alloy) Gold: 81.5, 14.5, 4 Silver: 58.5, 18.5, 23 |
| 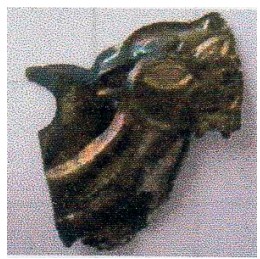 | PACEB-F4-00112 (gold and silver alloy) Gold: 81, 13.5, 5.5 Silver: 69, 25, 3 (3% As) |

\* The silver area is highly inhomogeneous, referred to in the content.

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
