# Peer review of "Studies and Considerations on Forty-Three Gold and Silver Nose Ornaments from the Moche Tomb of the Lady of Cao"

_heritage, doi:10.3390/heritage6090328_

Round 1

Reviewer 1 Report

In this work, the authors present a multi-technique analysis of 43 precious metals ornaments from the “Moche tomb of the Lady Cao”. Results aim to describe the objects from the material viewpoint and to provide insights on its manufacture. Applications of the techniques mentioned in this work can be considered as a rutinary procedure, the interest of the paper would be around the historical significance of the studied objects, and how important they are for the heritage of Peru. As it is now, I fail to find a proper archaeological description of the objects and the context where they were found, also it requires more emphasis on the description of the technology employed for the manufacture of the objects (which I would consider as the major contribution), especially by using the results generated in this work.

Please address the following issues prior to the publication of the paper:

ABSTRACT

Incorporate the analysis techniques within the text instead of presenting them as a list.

INTRODUCTION

Try to perform a review of similar previously published papers in “Heritage” journal, since it will help the reader to understand better which is the contribution of this work, and will put it in a more adequate “Heritage Sciences” context. Many of your bibliographical references come more from a technical point of view (physics of the analysis techniques, journals oriented towards natural sciences, etc.). Improve the bibliographical review, there are a significant number of publications dealing with the manufacture of gold/silver/copper alloys in diverse parts of the world, that describe different technological aspects.

Establish the actual contribution of this work and elaborate on the “Heritage significance” of the studied objects.

(lines 53-68) Improve the discussion on What’s new in this work, with respect to the previous one [ref.12].

Provide further information on the archaeological context. Establish the actual provenance of the objects. It would be nice if you provide pictures as well (context, general view of objects, etc).

DEPLETION GILDING

Can you provide better information on the chemical processes involved? Such as the chemical reactions? Is this process important for the preservation/conservation of the objects?

MATERIALS

As a recommendation, consider using a condensed ID code for the objects, in order to facilitate the reading. You could explain the meaning of the code, show the object+ID code in the supplementary materials and keep only the actual identifier of each object for the text.

RESULTS

(lines 148-149) You need to support this affirmation with previously published archaeological information. Or at least provide a brief discussion.

Figures 1 and 2. Improve definition of figures or generate better plots. Since, as they are now, seems to be digitalized from somewhere else.

DISCUSSION

(lines 184-185) Provide a more visual way to see this. Maybe a histogram or a scatter plot could help.

(lines 191-193) Maybe they are grouping? In figure 2 there are at least two possible groups of objects. Is there any archaeological information which could explain such differences? What kind of objects are in each cluster? On the other hand, have you considered a multi variable statistical analysis (on the elemental concentration)?

It might be better if instead of trying to prove wrong a previous author you try to establish an unbiased characterization of the objects, and then connect your results with the archaeological context. Finally, you could discuss your findings and contrat them with  previous theories regarding the technology involved in the object’s manufacture. If you follow this direction, it might be even clearer which is the actual contribution of your work.

(lines 217-228) You need to provide more information regarding these points. Provide information on the attenuation coefficient for the elements you are presenting, and the effective attenuation coefficient corresponding to the alloys you are observing. Is the thickness of the different regions comparable?

(lines 243-252) Provide evidences of each one of the methods you mention here. Or at least make it clearer where should the reader look for it in your figures.

(lines 243-245) I’m not sure if the term “button” is correct, by this you mean something like a rivet? Indicate in Figure 4, where this rivet is located. What kind of “glue” was employed? How can you determine its existence?

(lines 249-250) “…and possibly in other nose ornaments…” this statement lack of evidence. Remove it or provide proof of it.

(line 260) “…taking also into account their external aspect…” please explain the sentence. Have you considered to use microscopy of the surfaces as well? It could provide further information on the thermal treatments.

(261-310) I fail to see where you provide proof of the “archaeological interpretation of the data and a chronological reconstruction” that you mention. In my opinion this work should have a better archaeological basis, at least try to cite the previous archaeological works and interpretations, or the archaeological reports generated during the finding of the studied objects. There should be similar associated objects coming from similar archaeological contexts in which a proper chronology has been established (?) Usually identification of the workshops provides a dating by association as well.

(lines 289-290) How can you measure the “more “sophisticated” craftsmanship” that you mention? Establish the features that indicate this innovation.

(lines 298-299) The statement: “…which possibly refers to an object subjected to an unsuccessful depletion gilding process…” Have you considered possible weathering of the object? Where was it in the archaeological context from which has been extracted. If it was a failure, would it be offered as the other “more sophisticated and better achieved” pieces?

Perform an exhaustive review on syntax and grammar

Author Response

We would like to thank the kind referee for the work done 

Incorporate the analysis techniques within the text...  Done

Try to perform a review of similar previously published papers in “Heritage” journal....  We have added references to studies of gold artifacts in different contexts.

Many of your bibliographical references come more from a technical point of view...  We have added references to the El Brujo discovery, and to the study of gold artefacts in other regions .

Establish the actual contribution of.... the paper discusses the different aspects of the techniques used by the Moche to join together thin sheets of gold and silver. It is also shown that in almost all cases they used single sheets of gold and silver.

on the “Heritage significance” of the studied objects. We added references tothe Lady od Cao discovery.

(lines 53-68) Improve the discussion on What’s We add the following period in order to clarify our position:

"In this paper, results are presented which clearly show that the prevailing technique was to join separate gold and silver sheets. The results of the radiographs are very clear."

Provide further information on the archaeological context. .... We have added references to the El Brujo discovery ref. 5.

Can you provide better information on.... The chemistry of depletion gilding is partially unknown

Is depletion gilding important for the conservation of the objects? An object subject to depletion gilding (tumbaga) is better preserved than an object subject to gilding, because of the interface gold-copper, which is steep in the case of gilded copper, soft in the case of depletion gilding.

provide pictures of the context (general view of the objects) a picture of the mummy (fig.1) was added in the text

consider using a condensed ID code for the objects, in order to facilitate the reading. .... the common prefix PACEB-F4- was removed in the text and left in the supplementary materials

(lines 148-149) You need to support this affirmation with previously published archaeological information. The phrase was completely re-written, deleting any reference to a possible chronology, emphasizing only the fact that the significant differences between the objects implies that it took time to manufacture such different objects

Figures 1 and 2. Improve definition of figures or generate better plots. better plot were done

(lines 184-185) Provide a more visual way to see this. Maybe a histogram or a scatter plot could help.

(lines 217-228) : more information regarding these points  this section was changed as following:

"The transmission measurements resulted in significant differences in the attenuation of gold and silver areas, as expected if these areas actually were made on gold and silver alloys. As hypothesized by Ingo of gold subject to depletion gilding, both Ag and Cu present in the Au-alloy, would emigrate to the surface, being removed. In the Ag-areas, when only Cu would emigrate to the surface being removed. Finally, attenuation of gold and silver areas would be similar, starting from the same layer.  This is not confirmed from the transmission experiments, also given that the thickness of gold and silver alloys is comparable."

(lines 243-253)  : provide evidence of each one of the methods you mention here  

this section was partially rewritten as follows :

A  …….(see Figure 4a, showing two Ag-buttons for the two shields)

B……..(see Figure 5).  The soldering with Ag is demonstrated by the EDXRF-spectrum of the soldering area, showing an excess of Ag

C……..OK

D……. OK

(lines 249-250) : …….and possibly in other nose ornaments…..(lack of evidence)  This sentence was removed

(line 260) : ……taking into account their external aspect… Explain !

(lines 261-310) :  Proofs for the Archaeological interpretation of the data and chronological reconstruction The sentence was removed

(lines 289-290): more sophisticated craftsmanship ? From the aesthetic aspect, a  large number of nose ornaments appeared to be more refined and, therefore, possibly, of more recent production….

(lines 298-299) : have you considered the possible weathering of the object ?  which possibly refers to an object subjected to an unsuccessful depletion gilding process or  originally located in contact with the soil or with the body of the Lady of Cao…

Reviewer 2 Report

1. What is the main question addressed by the research? Analytical techniques used to study the metallurgy and techniques used in the preparation of silver and metal artifacts.    2. Do you consider the topic original or relevant in the field? Does it address a specific gap in the field? It does much expand knowledge of artifact manufacture.   3. What does it add to the subject area compared with other published material? It extends existing knowledge.   4. What specific improvements should the authors consider regarding the methodology? What further controls should be considered? I would recommend addition of XPS if possible.   5. Are the conclusions consistent with the evidence and arguments presented and do they address the main question posed? The conclusions and evidence are well matched.    6. Are the references appropriate? All references are appropriate.   7. Please include any additional comments on the tables and figures.

Tables and figures are well organized and informative.

Might the gap in the radiograph grey levels shown in Figure 9d be indicative of organic glue?

Author Response

We would like to thank the kind referee for the work done 

Might the gap in the radiograph grey levels (Figure 9d) indicative of organic glue?

No, the grey level only indicates a discontinuity between gold and silver x-RAY attenuation

Reviewer 3 Report

The article "STUDIES AND CONSIDERATIONS ON FORTY-THREE  GOLD AND SILVER NOSE ORNAMENTS FROM THE MOCHE TOMB OF THE LADY OF CAO
" describes new results on the manufacturing of ornaments from the tomb of Lady of Cao, IV-V CE, Peru.

The aim of the study is clear, however, some statements - in my opinion - have  to be revised and/or integrated.

Suggestions are in the following:

Please unify the notation of the calendar along the article.

Please use the correct font in the article.

Please, correct the typos.

Please provide further specifications on the three set-up (EDXRF, XRT, RAD) used for the study, e.g. source and detector used.

Please provide at least one EDXRF spectrum of the data presented in Tab.1, for  better understanding the acquired data and how they are processed for the analysis. For example, report the experimental uncertainty for each element in the measurement, and discuss why it is not relevant in the analysis.

Please, underline if the absolute or relative quantification in achieved with the anaysis. 

Please, specify the importance of the study of the manufacturing of the ornaments in the context of the study of the tomb of Lady of Cao.

In line 96, please specify which one.

In Fig.1, report in the label who is indicated with the red and black dots. Moreover, is quite impossible for me to distinguish the points in the plot.

In Fig.3, please describe the graph in the label.

Author Response

We would like to thank the kind referee for the work done.

Unify the notation of the calendar, use the correct font and correct the typos done

Provide further specifications on the three set-up (EDXRF, XRT, RAD), e.g. detector used see new line 145

Please provide one EDXRF spectrum of the data present in Table 1  OK, see new Figure 4

Report and discuss the experimental uncertainty The following two periods were added.

The total uncertainty associated to the EDXRF experimental measurements  of  gold – areas containing Au, Ag and Cu-concentration, is depending on various reasons, such as the manufacturing method,  the restoration  treatment, the analytical method (EDXRF),  the analysis of the EDXRF-spectrum, and the reference samples employed for EDXRF-analysis . All these contributions are discussed in detail in par. 3.1.5 of Ref. 5.

The total uncertainty associated to the EDXRF experimental measurements  of  silver – areas containing  Ag , Cu and Au concentration, is much more critical, mainly depending on the surface enrichment processes at the surface

Please, underline if the absolute or relative quantification in achieved with the anaysis. The quantification was done using different standards of gold and silver alloys. So it's absolute.

in line 96, specify which one ??

In Figure 1, indicate the black and red dots the two figure were re-plotted and red circle eliminated

Please, specify the importance of the study of the manufacturing of the ornaments in the context of the study of the tomb of Lady of Cao. those made in recent years are the first systematic analyzes of this funerary equipment discovered in 1987.

In Fig.3, please describe the graph in the label. we have added the following proposition "…….(top figure), its radiography and the grey level profile of the interface gold-silver. A clear difference may be observed……."

Round 2

Reviewer 1 Report

In my opinion there is still lack of information on the depletion gilding process and how it could influence the conservation of the objects. But as the paper is now, it seems that it would not affect the main conclusions.

I do not see any contribution of figure 2. Maybe a diagram of the apparatus could work better?

Perform a general check for typos

Author Response

The figure 2 has been changed. The depletion process is really unknown,it can be understood in principle but not in the details, which are essential for a real understanding of the technique.

Thank you again

Reviewer 3 Report

I thank the authors for having accepted the comments.

Author Response

Many thanks to you.